# Metamorphic Forward Adaptation Network: Dynamically Adaptive and Modular Multi-layer Learning

**Yu Sun**                                                                          *sofia.sun@berkeley.edu*
*Department of Computing, Imperial College London*

**Vijja Wichitwechkarn**                                                            *vw273@cam.ac.uk*
*Department of Engineering, University of Cambridge*

**Ronald Clark**                                                                    *ronald.clark@cs.ox.ac.uk*
*Department of Computer Science, University of Oxford*

**Mirko Kovac**[1,2,3]                                                              *mirko.kovac@empa.ch*
[1]*Laboratory of Sustainability Robotics, Swiss Federal Laboratories for Materials Science and Technology (Empa)*
[2]*Laboratory of Sustainability Robotics, École Polytechnique Fédérale de Lausanne (EPFL)*
[3]*Aerial Robotics Laboratory, Department of Aeronautics, Imperial College London*

**Basaran Bahadir Kocer**                                                           *b.kocer@bristol.ac.uk*
*School of Civil, Aerospace and Design Engineering, University of Bristol*

**Reviewed on OpenReview:** *https://openreview.net/forum?id=6RCs2tLsHq&noteId=53TwP6wCPO*

## Abstract

Back-propagation is a widely used algorithm for training neural networks by adjusting weights based on error gradients. However, back-propagation is biologically implausible with global derivative computation and lacks robustness in long-term dynamic learning. A previously proposed alternative to back-propagation is the Forward-Forward algorithm, which bypasses global gradient dependency and localises computations, making it a more biologically plausible approach. However, Forward-Forward has been evaluated in limited environments, does not yet match back-propagation's performance, and only supports classification, not regression. This research introduces the Metamorphic Forward Adaptation Network (MFAN), using a contrastive learning property as its core, and retaining the layer-wise architecture of the Forward-Forward algorithm. Compared to the Forward-Forward model being limited to discrete classification, MFAN can process discrete and continuous data, showing stability, adaptability, and the ability to handle evolving data. MFAN performs well in continuous data stream scenarios, demonstrating superior adaptability and robustness compared to back-propagation, particularly in tasks requiring dynamic, long-term learning.

## 1 Introduction

Deep learning has achieved remarkable success across numerous applications, primarily due to the efficiency of the back-propagation (BP) algorithm in training deep neural networks. BP has been the foundation for advancements in fields ranging from image recognition to natural language processing. However, as these models are increasingly deployed in dynamic environments, such as adaptive systems and continuous data stream scenarios, the limitations of BP become more apparent.

One notable limitation of BP is the weight symmetry requirement (the weight transport problem), which necessitates that the weights for forward and backward propagation remain identical throughout the training process (Bengio et al., 2015). Additionally, BP requires distinct inference and learning phases, with error sig-

nals propagated backward layer by layer. This process depends on globally synchronised updates and precise knowledge of activation function derivatives, both of which are biologically implausible. Alternatives such as Feedback Alignment (FA) and Direct Feedback Alignment (DFA) address these constraints by replacing exact weight symmetry with fixed random feedback weights or direct error signals by offering mechanisms that align more closely with biological plausibility (Nøkland, 2016). Furthermore, recent studies of wide neural networks and learning rules in the Neural Tangent Kernel (NTK) regime show that simpler methods can achieve comparable performance by leveraging local error-weighted input correlations instead of relying on global feedback synchronisation (Boopathy & Fiete, 2022). For a more detailed discussion of biologically plausible methods, readers are referred to (Lv et al., 2024).

Another significant shortcoming of BP is its limitations in real-time applications requiring sustained plasticity (Dohare et al., 2024). BP relies on precise knowledge of forward-pass computations to calculate accurate gradients, which can result in ineffective gradient updates in scenarios where underlying data distributions are constantly changing. This dependence on static computations hinders BP's ability to adapt to dynamic environments, as gradient updates derived from a batch of training data may fail to generalize effectively to evolving tasks. Consequently, BP-trained models often struggle to adapt or generalize in such settings.

As a solution, adaptive neural networks (Ashfahani & Pratama, 2019), which dynamically modify their architecture by expanding or contracting based on task complexity, have demonstrated superior performance in real-time settings (Kocer et al., 2021). For example, tasks involving continuously streaming data demand immediate adaptation and stable predictions, where adaptive models have proven to be more effective. While there is some evidence of the backward flow of error signals in biological systems, which is central to BP (Francioni et al., 2023), neurons in the brain are believed to operate primarily through local updates rather than the global error propagation characteristic of BP. This disconnect has driven interest in exploring alternative learning methods (Karimi et al., 2024; Lv et al., 2024), such as the Forward-Forward (FF) algorithm (Hinton, 2022).

The FF algorithm is a greedy multi-layer learning procedure inspired by Boltzmann machines (Hinton et al., 1986) and Noise Contrastive Estimation (Gutmann & Hyvärinen, 2010). FF replaces BP's forward-backward passes with two forward passes – one with positive (real) data and one with negative (generated or corrupted) data. Each layer of FF has its local objective function aiming to have high goodness for positive data and low goodness for negative data through weight adjustments in every hidden layer, as illustrated in Fig. 1a.

An example of a goodness measure is the sum of the squares of the activities of the rectified linear neurons in each layer, aiming to learn to make the positive data goodness above a threshold $\theta$ and the negative data below. Thus FF can classify input vectors by computing the probability that an input vector is positive. This is achieved by applying the logistic function, $\sigma$, to the difference of the goodness and $\theta$,

$$p(\text{positive}) = \sigma \left( \sum_j y_j^2 - \theta \right) \tag{1}$$

where $y_j$ is the activity of hidden unit $j$ before layer normalization.

While BP requires a fully differentiable forward process to propagate learning signals throughout the network, FF can free this constraint by having each layer learn with locally generated loss signals.

Even though FF provides a more modular, layer-wise learning approach, it tends to be slower, particularly when applied to large-scale tasks, and its reliance on local optimisation can result in sub-optimal coordination across layers, potentially reducing overall performance in larger networks.

Researchers have explored several enhancements to the FF algorithm, including hyper-parameter optimisation and hybrid approaches (Gandhi et al., 2023; Izzo et al., 2024; Kumar et al., 2023; Ahamed et al., 2023; Reyes-Angulo & Paheding, 2024; Wang et al., 2024). One study (Gandhi et al., 2023) introduced a pyramidal strategy for adjusting the loss threshold progressively across layers, enhancing performance on tasks beyond MNIST classification, including sentiment analysis on the IMDb dataset. Additionally, integrating the FF algorithm with BP (Izzo et al., 2024) has shown promising results, combining the robustness of BP with the biological plausibility of FF. These hybrid models have demonstrated improved noise resilience and adaptability, especially in environments where task complexity evolves.

In light of the efforts and limitations outlined above, our research identifies an additional constraint of the FF model that has yet to be addressed. The FF model's design combines both the feature vector ($x$) and the target variable ($y$) as inputs. During inference, this requires knowledge of all possible discrete $y$ values to combine with the test $x$ to create valid test inputs. The model then computes goodness values for these inputs and selects the $y$ prediction corresponding to the highest goodness value. While this design facilitates classification, it poses challenges for regression tasks.

To address these limitations, this paper proposes the Metamorphic Forward Adaptation Network (MFAN), a novel architecture using contrastive learning property as its core, and utilising a layer-wise architecture similar to FF. MFAN extends FF's algorithm to the continuous data domain, supporting both regression and classification.

## 2 The Proposed Model

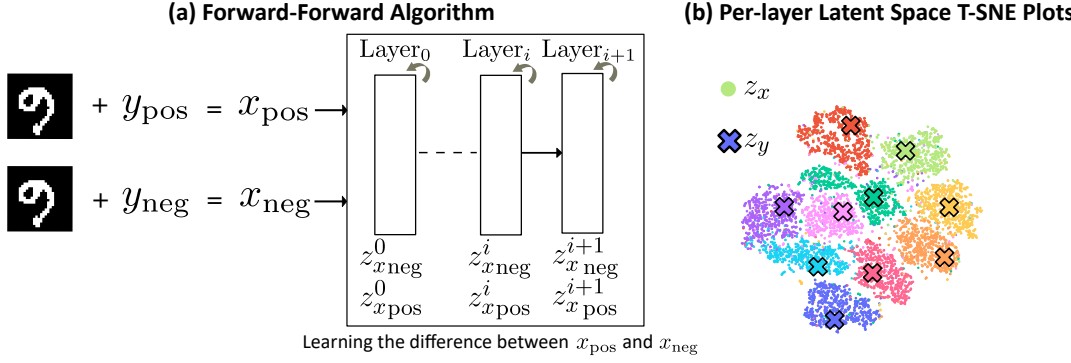

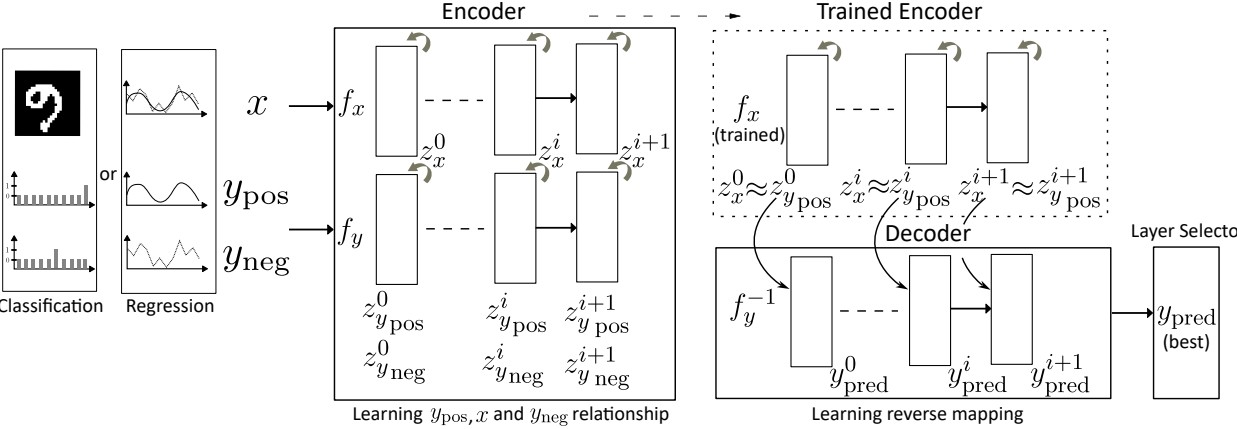

Figure 1: **a)** Schematic for the Forward-Forward (FF) model. The FF model combines x data and y labels into "good" and "bad" data inputs. Each layer's loss function maximises/minimises goodness for 'good'/'bad' data. At inference, all labels are used, and the label with the highest goodness is taken as the output. Hence, FF can not be applied to continuous data, such as regression. **(b)** Latent space embedding for each MFAN layer in MNIST classification tasks. **c)** Schematic for the proposed approach: MFAN consists of an input encoder $f_x$, output encoder $f_y$ and a decoder $f_y^{-1}$. The encoder maps the input data into the same latent space, and the Cosine Similarity contrastive loss is used to pull positive samples of $x$ and $y$ together and push negative samples apart. The decoder maps from the latent space to the predicted output. Cross-entropy or Root Mean Square loss can be used to learn the reverse mapping. The effective layer selector selects the layer that makes the final prediction.

The first improvement in the design of MFAN and the FF algorithm is data processing. Negative $y$ ($y_{\text{neg}}$) is generated from positive $y$ ($y_{\text{pos}}$) by either choosing the wrong label (classification) or introducing random noise offsets (regression). MFAN no longer uses combined $x_{\text{pos}}$ $x_{\text{neg}}$ as input, and instead separately encodes data and its positive and negative labels into the same $d$-dimensional latent space with an input encoder $f_x$, and an output encoder $f_y$

$$z_x = f_x(x) \tag{2}$$
$$z_{y_{\text{pos}}} = f_y(y_{\text{pos}}) \tag{3}$$
$$z_{y_{\text{neg}}} = f_y(y_{\text{neg}}) \tag{4}$$

obtaining $z_x, z_{y_{\text{pos}}}, z_{y_{\text{neg}}} \in \mathbb{R}^d$.

Then, replacing FF's goodness measure, each MFAN layer independently optimises a contrastive loss using cosine similarity (Lahitani et al., 2016). Cosine similarity is a metric used to measure the similarity between two vectors by calculating the cosine of the angle between them,

$$\text{cos\_sim}(v_1, v_2) = \frac{v_1 \cdot v_2}{\|v_1\|\|v_2\|}. \tag{5}$$

As cosine similarity measures the alignment between vectors, the contrastive loss function uses this property to encourage MFAN to learn to pull $z_x$ and $z_{y_{\text{pos}}}$ together and push $z_x$ and $z_{y_{\text{neg}}}$ apart in latent space $\mathbb{R}^d$,

$$\mathcal{L}_{\text{cos\_sim}} = -\frac{z_x \cdot z_{y_{\text{pos}}}}{\|z_x\|\|z_{y_{\text{pos}}}\|} + \alpha_1 \frac{z_x \cdot z_{y_{\text{neg}}}}{\|z_x\|\|z_{y_{\text{neg}}}\|} + \alpha_2 \frac{z_{y_{\text{pos}}} \cdot z_{y_{\text{neg}}}}{\|z_{y_{\text{pos}}}\|\|z_{y_{\text{neg}}}\|} \tag{6}$$

where $\alpha_1$ and $\alpha_2$ are tunable parameters, the last term of this loss aims to retain the distance between positive and negative $y$ data mappings.

Using this loss function, MFAN encoders are first trained together to obtain mappings from the input/output space to the latent space, pulling $z_x \simeq z_{y_{\text{pos}}}$. The encoder parameters are then frozen and used to obtain $(y, z_y)$ pairs. These are used to train a decoder $f_y^{-1}$, which learns the reverse mapping of $z_{y_{\text{pos}}}$ back to $y_{\text{pos}}$. As the loss minimisation of encoder maps $z_x$ and $z_{y_{\text{pos}}}$ together through training as shown in Equation 6, then this condition is also expected to hold in the test set at inference.

$$\hat{y}_{\text{test}} = f_y^{-1}(f_x(x_{\text{test}})) \simeq g_y(z_{x_{\text{test}}}). \tag{7}$$

The input encoder and decoder are used together to make predictions as detailed in Algorithm 1. Thus, MFAN can predict without knowing all possible $y$, hence being able to process continuous data and do regression tasks that FF is unable to perform.

Each layer in MFAN makes a prediction, and an effective layer selector determines which layer to use for the final output as detailed in Algorithm 2. For classification tasks, the layer with the best training performance is used during inference, or multiple top-performing layers can be averaged for smoother and more stable predictions. For regression tasks, the selector identifies the best-performing layer during training and uses a deeper layer at inference to address task complexity and extrapolation. With a continuous data stream, the model dynamically adjusts depth by finding the best-performing layer from the previous time step and using that layer at the subsequent inference time. The proposed approach is illustrated in Fig. 1c.

---

**Algorithm 1** MFAN Pseudo-code (Encoder & Decoder)

---

1: **Input:** digits $x$, correct labels $y_{\text{pos}}$, wrong labels $y_{\text{neg}}$
2: **for** Encoder layer $L_i$ **do**
3:   $z_x^i \leftarrow f_x^i(z_x^{i-1})$
4:   $z_{y_{\text{pos}}}^i \leftarrow f_y^i(z_{y_{\text{pos}}}^{i-1})$
5:   $z_{y_{\text{neg}}}^i \leftarrow f_y^i(z_{y_{\text{neg}}}^{i-1})$
6:   $\mathcal{L}_{\text{cos\_sim}}^i = -\frac{z_x^i \cdot z_{y_{\text{pos}}}^i}{\|z_x^i\|\|z_{y_{\text{pos}}}^i\|} + \alpha_1 \frac{z_x^i \cdot z_{y_{\text{neg}}}^i}{\|z_x^i\|\|z_{y_{\text{neg}}}^i\|} + \alpha_2 \frac{z_{y_{\text{pos}}}^i \cdot z_{y_{\text{neg}}}^i}{\|z_{y_{\text{pos}}}^i\|\|z_{y_{\text{neg}}}^i\|}$   $\triangleright$ Compute layer-wise cosine-similarity loss
7:   $w^i \leftarrow w^i - \eta \, \nabla_{w^i}\big(\mathcal{L}_{\text{cos\_sim}}^i\big)$        $\triangleright$ Local weight update
8:   Apply normalization and ReLU activation to $z_x^i$, $z_{y_{\text{pos}}}^i$, $z_{y_{\text{neg}}}^i$, pass to the next layer
9: **end for**
10: **for** Decoder layer $L_i$ **do**
11:   $\hat{y} \leftarrow g_y^i(z_x^i)$       $\triangleright$ Map intermediate digit representation back to label space
12:   $\mathcal{L}_{\text{CE}}(\hat{y}, y_{\text{pos}}) = -\sum_c y_{\text{pos},c} \log(\hat{y}_c)$      $\triangleright$ Compute cross-entropy loss
13:   $w^i \leftarrow w^i - \eta \, \nabla_{w^i}\big(\mathcal{L}_{\text{CE}}\big)$        $\triangleright$ Local weight update
14: **end for**
15: **Output:** Predicted labels

---

**Algorithm 2** MFAN Pseudo-code: Layer Selector

---

1: **Input:**
  $\{L_1, L_2, \ldots, L_N\}$: Trained MFAN layers
  $D_{\text{train}}$: Training (or validation) dataset
  $\mathcal{M}$: Performance metric (e.g., accuracy or MSE)
2: **Initialize:** $perf[i] \leftarrow 0$, $i = 1, \ldots, N$
3: **for each epoch do**
4:   **for** $i = 1$ **to** $N$ **do**
5:    $perf[i] \mathrel{+}= \mathcal{M}\big(L_i, D_{\text{train}}\big)$
6:   **end for**
7: **end for**
8: $i^\star \leftarrow \arg\max_i\{perf[i]\}$
9: **Output:** $i^\star$

---

# 3 Experiments and Evaluations

## 3.1 Classification Evaluations

This section evaluates the ability of MFAN to handle standard classification tasks and compares its performance to the FF and BP models. While FF and BP have been extensively tested on classification tasks in prior research, with BP consistently outperforming FF, MFAN was primarily designed to address a different limitation of FF: its unsuitability for regression tasks. No prior work has applied FF to regression problems to the best of our knowledge, as it has only been evaluated on classification tasks. Here, we aim to demonstrate that MFAN can also perform classification tasks, doing better than FF but still falling short of BP's performance. These results help establish a baseline for MFAN's classification capabilities before focusing on its primary advantages in regression tasks in dynamic environments.

### 3.1.1 MNIST Classification: Performance on Digit Classification

The MNIST dataset was used to evaluate MFAN on a simpler classification task and compare its performance to FF and BP. The configurations of all models were designed for comparability rather than peak optimisation. The model setup configurations are given in Appendix A.1.1.

MFAN, FF, and BP employ fundamentally different learning mechanisms. BP learns the relationship between MNIST digits and their labels, FF distinguishes positive and negative data within each layer based on a unique processing structure, while MFAN not only learns the digit-label relationship but also captures why a digit is not associated with other labels through its contrastive learning framework. These differences influence computational demands and the resulting test accuracy: BP achieved the best performance in terms of both accuracy and training efficiency, with an average test accuracy of 94.36%±0.46 and a training time of 5 seconds. MFAN, while trailing BP by 2.71% in accuracy (91.65% ± 0.20%), demonstrated more consistent performance, albeit with a longer training time of 33 seconds due to its more complex computations. FF struggled significantly in its default setup, achieving only 45% test accuracy after 42 seconds of training. To reach 90.5% ± 1.42%, FF required much wider networks with nearly 10 times more iterations, extending its training time to 375 seconds. These results highlight MFAN's ability to outperform FF in terms of both accuracy and stability, while still falling short of BP in this classification task.

These results highlight MFAN's ability to learn meaningful representations despite its lower accuracy compared to BP. This is further supported by a t-SNE visualisation of the latent space learned by MFAN (Fig. 1b), which revealed well-defined clusters of digit labels, demonstrating each of its layers can capture semantically meaningful features.

### 3.1.2 CIFAR-10 Classification: Performance on Image Classification

The CIFAR-10 dataset was used to evaluate MFAN, BP, and FF on a more challenging image classification task. Fully connected architectures were employed rather than convolutional layers for comparison purposes and to maintain consistency with other tests in the paper. Previous research ((Lin et al., 2015)) has shown that pure BP models achieve around 51% test accuracy on CIFAR-10 with fully connected layers, providing a key reference point for our comparisons. The model setup configurations are given in Appendix A.1.2.

In the short training-time setup, BP again demonstrated superior performance, achieving an average test accuracy of 50.72%±0.21% with a training time of 21 seconds. MFAN reached 37.47%±1.38% test accuracy in 25 seconds, while FF achieved only 13.41% ± 2.92% after 30 seconds of training. However, with wider network architectures – keeping the same 3-layer, short-training setup but scaling each layer to 10 times the number of nodes, from hundreds to thousands – and longer training times, both FF and MFAN improved, achieving test accuracies reaching 46%. Nonetheless, their performance is still inferior to that of the BP model, with also 10 times more nodes, which achieved 49% test accuracy. In all these conditions, we compared them without additional tuning and only increased the network width by adding 10 × more nodes to each layer. In such a case, we observed overfitting in BP and some performance increases as discussed for FF and MFAN.

It is worth noting that while MFAN and FF models improved their accuracy with 10 times wider layers, they came at the cost of extended training times, leading to a total of approximately 30 minutes for both MFAN and FF models. This made further optimisation impractical and not worthwhile. In contrast, the BP model with 10 times more nodes required only around 2 minutes for training, underscoring the relative time inefficiency of the FF and MFAN models in vision tasks like CIFAR-10 classification as compared to BP. Overall, BP remained the most efficient and accurate model for this classification task.

Table 1: Classification results: Test accuracy.

| Dataset | MFAN | BP | FF[1] |
|---|---|---|---|
| MNIST | 91.65% ± 0.20% | 94.36% ± 0.46% | 90.50% ± 1.42% |
| CIFAR-10 | 37.47% ± 1.38% | 50.72% ± 0.21% | 13.41% ± 2.92% |

[1]The best performance with short training time.

**Discussion: Moving Beyond Classification**

The results from the MNIST and CIFAR-10 evaluations confirm that MFAN is capable of performing classification tasks and outperforms FF in this domain. However, it falls short of BP, which remains more efficient and better suited for standard classification benchmarks. This outcome is consistent with MFAN's design

focus, as it was developed primarily to address regression and dynamic data challenges where FF and BP have limitations.

Unlike BP and MFAN, FF cannot perform regression tasks or handle continuous data. Its design, centred on measuring goodness for each labelled data, inherently limits its ability to model smooth functional relationships required for regression. These design constraints make FF unsuitable for tasks involving continuous data streams or dynamic environments, where adaptability and precise representation of input-output relationships are critical.

MFAN, on the other hand, is specifically designed to address these challenges. Its layer-wise contrastive learning framework, dynamic depth adjustment, and robustness to noise make it better suited for handling short, dynamic data streams and evolving data distributions. The following sections will focus on MFAN's primary strength: its ability to perform regression tasks in dynamic environments and its advantages over BP and FF in these scenarios.

### 3.2 Function Regression: Continuous Data Learning

### 3.2.1 Evaluation Setup

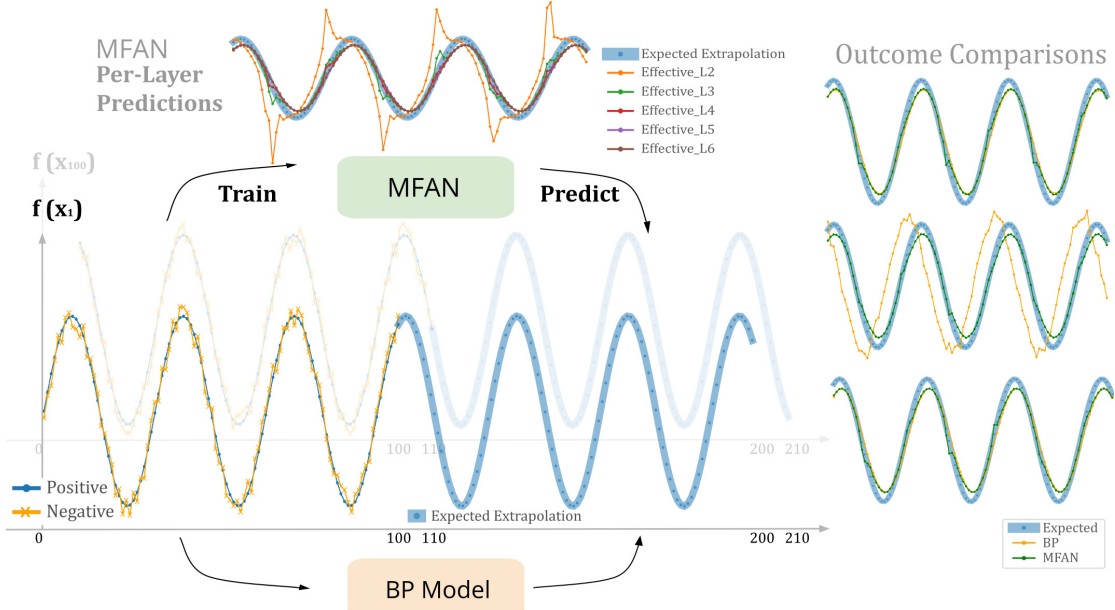

Figure 2: Schematic for function regression extrapolation task procedure. Positive data pairs, represented by the thin blue line, consist of $x \in \mathbb{R}^d, y_{\text{pos}} = f(x) \in \mathbb{R}^d$ where $f$ is the target function and $d = 100$. Negative data pairs, represented by the thin orange line, consist of $x \in \mathbb{R}^d, y_{\text{neg}} = y_{\text{pos}} + \vec{r} \in \mathbb{R}^d$ where $\vec{r}$ as a random offset. The expected extrapolation data $(x_{\text{test}}, y_{\text{test}}) \in (\mathbb{R}^d, \mathbb{R}^d)$ in shown by the thick blue line. Some MFAN and BP performance comparison plots are shown.

We evaluate MFAN on continuous data using a series of trigonometric functions as regression targets (Hay & Sharon, 2024; Xu et al., 2021). We use a three-layer BP model as the baseline for comparison, and we detail the model configuration in Appendix A.2. To explore the adaptive layer selection mechanism of MFAN, we implement a six-layer MFAN architecture while also evaluating both models with the same number of layers for fairness, as detailed in Appendix A.2.1. The six-layer setup allows MFAN to demonstrate dynamic depth adjustment, which is not fully observable in the three-layer configuration, as it can only shrink the depth but cannot expand it. In fact, MFAN uses three layers most frequently during inference, similar to BP, but its dynamic depth adjustment enables more flexible predictions.

The experimental setup involves constructing datasets using a sliding window approach. Specifically, the training dataset spans the domain $[0, 110]$, with data points $x \in \mathbb{R}^d, y_{\text{pos}} = f(x) \in \mathbb{R}^d, d = 100$, and

$x_{i+1} = x_i + \mathbf{0.\vec{1}}i$ with $x_0 = \vec{\mathbf{0}}, i \in \mathbb{Z}$, and $i \in [1, 100]$. Negative data $y_{\text{neg}}$ are generated by adding random offsets to $y_{\text{pos}}$, creating $y_{\text{neg}} = y_{\text{pos}} + \vec{\mathbf{r}}$. The test dataset spans the domain $[100, 210]$ and evaluates extrapolation performance. The training and evaluation processes are repeated to obtain average results, which are analysed to assess both accuracy and stability across all evaluations. Figure 2 visualises the first and last sets of data pairs for the training and testing datasets.

### 3.2.2 Results & Analysis

Observations show that MFAN dynamically chooses output layers during inference, adjusting the network depth based on task requirements. On average, MFAN uses three layers during inference, matching BP's fixed-depth setup. However, when configured with six layers, MFAN demonstrated the ability to expand or shrink its effective depth, providing more prediction choices than BP. This flexibility is particularly evident in more complex regression tasks, where deeper layers are utilised for extrapolation to achieve smoother and more accurate predictions. Conversely, BP's fixed architecture limits its adaptability, resulting in less accurate and less stable prediction plots. Both models exhibited comparable inference times, averaging between 0.03 and 0.06 seconds per evaluation, indicating that MFAN's dynamic architecture does not compromise prediction efficiency.

Sensitivity analysis further highlights MFAN's robustness. We introduced Gaussian noise with various standard deviation levels (ranging from 0.5 to 20) during training. With small standard deviations (0.5 to 5), MFAN's error rate is observed to increase by around 2%, while BP's error rate rises by 8%. With high noise levels (standard deviation range from 5 to 20), MFAN maintained a relatively steady performance, with a 6% error rate increase, whereas BP's error rate rose rapidly, leading to over 400% error rate increase and poor predictions overtaken by the added noise. This superior noise tolerance is likely due to MFAN's embedded noise design, where the model intentionally introduces noise in $y_{\text{neg}}$ during training and learns to distinguish it from $(x, y_{\text{pos}})$ pairs. This mechanism enhances MFAN's ability to handle noisy data more effectively than BP.

Table 2 summarises the regression performance metrics, presenting the average percentage errors and standard deviations for both MFAN and BP across these function regression tasks. MFAN has much lower standard deviation values and has not been observed to fail any regression tasks in this experiment. The average percentage root mean square errors (RMSE) of MFAN are also lower than BP. These results demonstrated that MFAN consistently outperformed BP in stability and accuracy, emphasising its ability to handle complex, nonlinear functional relationships and extrapolate effectively to unseen domains.

These findings highlight MFAN's robustness in handling continuous data, particularly in scenarios where the relationship between input and output variables is complex and nonlinear. Its combination of dynamic depth selection, contrastive learning principles, and layer-wise weight updates makes MFAN a powerful tool for function regression tasks. Furthermore, MFAN's demonstrated resilience to noise and adaptive architecture further validates its suitability for scenarios requiring high stability, accuracy, and the ability to handle variability effectively.

Table 2: Regression results: Percentage mean $\pm$ standard deviation.

| Regression on $f(x)$ | MFAN[2] error | BP error |
|---|---|---|
| $f(x) = \sin(0.2x)$ | $06.43\% \pm 0.22\%$ | $10.59\% \pm 04.02\%$[1] |
| $f(x) = \cos(0.2x)$ | $06.41\% \pm 0.35\%$ | $21.16\% \pm 10.11\%$[1] |
| $f(x) = \sin(0.4\cos(0.2x))$ | $07.09\% \pm 1.09\%$ | $19.63\% \pm 12.13\%$[1] |
| $f(x) = \sin(0.4\cos(0.2x) + 30)$ | $10.39\% \pm 0.07\%$ | $13.48\% \pm 00.56\%$ |
| $f(x) = \sin(0.8\cos(0.2x) + 20)$ | $09.03\% \pm 0.11\%$ | $18.51\% \pm 11.68\%$[1] |
| $f(x) = \cos(0.4\sin(0.2x) + 3)$ | $12.26\% \pm 2.12\%$ | $13.23\% \pm 02.30\%$ |

[1]High standard deviation values in BP results indicate unstable predictions, leading to significant performance fluctuations.
[2]MFAN's stability and lower error rates consistently outperform BP across all evaluations.

### 3.3  Continuous Data Stream: Adapting to Dynamic Environments

#### 3.3.1  Evaluation setup

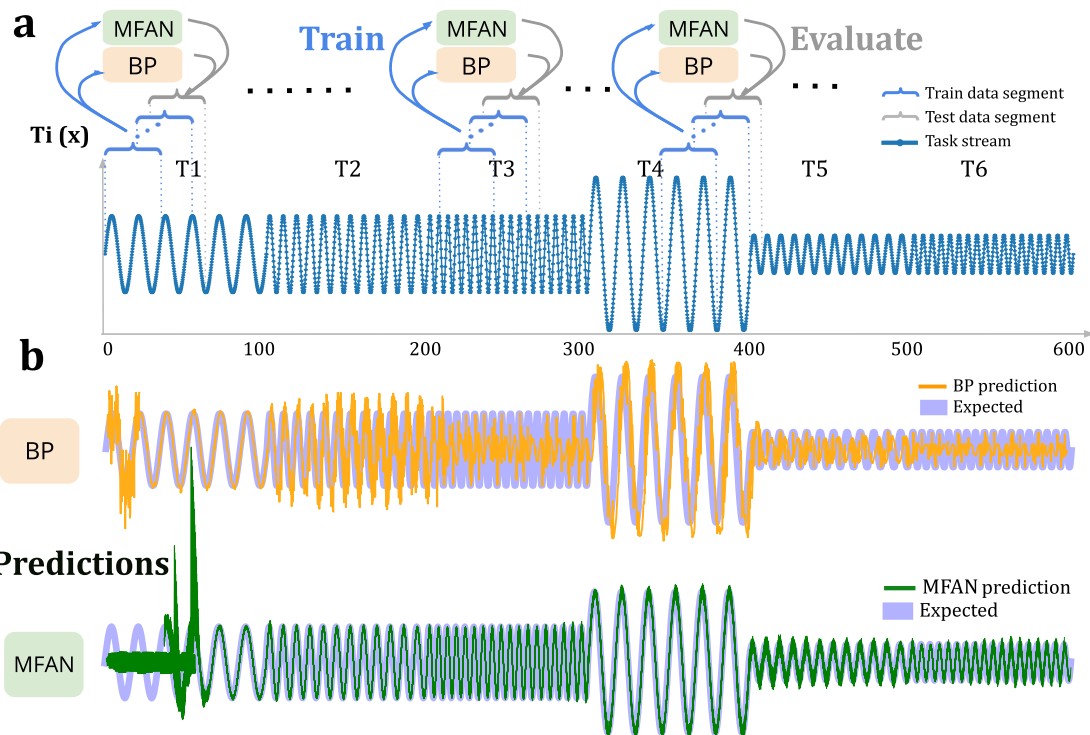

Figure 3: **a)** The evaluation setup for continuous data stream: training and evaluation process involves six sinusoidal task variations ($T_i$), each spanning a domain of length $30\pi$. These tasks are concatenated continuously to form a smooth task stream, divided evenly into 3000 points along the x-axis. Data segments are constructed using a sliding window approach, with each segment representing a vector in $\mathbb{R}^{100}$. The model is trained on five consecutive data segments and tested on the subsequent segment. This procedure is repeated iteratively until all points in the task stream have been processed. **b)** Prediction plots: the predictions from BP and MFAN models during the evaluation phases are plotted against the expected values for the three task streams.

This set of experiments evaluates MFAN's performance in continuous data stream tasks with model configuration details in Appendix A.3, where the model is required to learn and adapt to a sequence of evolving trigonometric functions (Fig. 3). These tasks simulate dynamic environments where the data distribution changes over time, forcing the model to continuously update its knowledge and adapt to new, unseen data. We use six tasks, each represented by a function defined in a domain of length $30\pi$. These tasks are concatenated continuously to form a smooth task stream (TS), which is then divided into 3000 evenly spaced points ($p_i$). These points are grouped into 100-dimensional data segments using a sliding window approach: $\text{data}_1 = [p_1, ..., p_{100}], \text{data}_2 = [p_2, ..., p_{101}]$, and so on until the final segment $\text{data}_{2901} = [p_{2901}, ..., p_{3000}]$.

The training and evaluation process is structured such that the model is trained on five consecutive segments and then tested on the next segment in the sequence. This procedure is repeated iteratively until all data points in the task stream have been processed. For the schematic of the continuous data stream evaluation process, refer to Fig. 3. Three task streams (TS1, TS2, and TS3) were designed to evaluate MFAN's ability to handle task transitions, adapt to unseen data, process complex patterns, and reorganise after interruptions.

#### 3.3.2  Results & Analysis

- **Sinusoidal Function Series (TS1)**

TS1 involved sinusoidal functions with varying frequency and amplitude, testing the model's adaptation ability to the smooth and nonlinear changes (Fig. 4a-TS1). MFAN initially learns the first task slowly but adapts well to task transitions and new data, producing stable and accurate predictions. In contrast, BP learns the first task quickly but struggles to adapt to subsequent tasks. The error rate plots (Fig. 4b-TS1) indicate that MFAN frequently adjusted its layers at the beginning of TS1. Per-layer error plots (Fig. 4c) suggest that at the start, its shallow layer ($L1$) had random predictions with very high errors, while its deeper layers ($L2$, $L3$) produced conservative predictions due to the lack of meaningful directional information from shallow layers. As $L1$ learned discriminating features, it makes better predictions than deeper layers, which seems to overfit. The effective layer selector in MFAN is adjusted to select the best-performing layer during training, thereby reducing error rates and stabilising performance. In contrast, BP showed sharp increases in error and fluctuations, indicating poor adaptation.

In terms of computational performance, both MFAN and BP demonstrated impressive efficiency. The training and inference times for both models were in the millisecond range (Table A5), with MFAN's training taking slightly longer than BP's, but still within a very fast timeframe. Despite this, MFAN's superior ability to adapt to evolving tasks and maintain stability across long-term learning cycles presents a significant advantage in dynamic environments.

- **Trigonometric Function Series (TS2)**

TS2 featured a broader range of trigonometric functions, requiring the model to capture more complex relationships and more intricate patterns (Fig. 4a-TS2). MFAN maintained a stable error rate throughout, efficiently handling the increased pattern complexity and variety. In contrast, BP's error rates became increasingly unstable during task transactions (Fig. 4b-TS2).

- **Linear and Trigonometric Function Series (TS3)**

TS3 introduced a linear function amid a series of trigonometric tasks to test the models' reaction to sudden drastic changes in data distribution (Fig. 4a-TS3). This "injury" task, where a linear function interrupts the periodicity of trigonometric functions, disrupted the learning process. MFAN struggled during "injury", attempting to maintain periodical predictions. Multiple layer adjustments were observed as MFAN actively attempts to recognise the "injury" (Fig. 4b-TS3). This also proves that the MFAN learning procedure has been hindered in this period. Once the task switched back to trigonometric functions, MFAN quickly adapted and demonstrated high accuracy. This ability to recover and re-establish stable predictions after "injury" illustrates MFAN's capability to recover in continuous data stream scenarios. In contrast, the BP quickly picked up the linear behaviour of "injury". Its reliance on gradient-based optimisation allowed it to learn the linear relationship and reduce error rates quickly. However, BP suffered from deteriorated performance when shifting back to nonlinear tasks.

These results highlight MFAN's robustness in handling dynamic, real-world environments where the ability to adapt to evolving tasks is critical. MFAN's layered learning approach allows it to selectively focus on different aspects of the data, maintaining stability and performance when faced with evolving tasks. MFAN also demonstrated recovery when the data stream evolved from a new phase to an older pattern, which is worth further investigation. This investigation would involve the biological plausibility of the proposed approach.

During the 'injury' period in TS3, MFAN appears to struggle as it attempts to maintain periodical predictions. Multiple layer adjustments were observed, reflecting the model's active efforts to recognise the 'injury.' Despite this, MFAN quickly adapted and demonstrated high accuracy once the task returned to trigonometric functions. Additional analysis reveals that MFAN decreases its error more effectively and becomes more stable during longer periods of consistent patterns. With 30% shorter data points, MFAN receives less consistent patterns each time, having 2% to 3% larger error rate amplitudes with oscillatory behaviour. Then, with 20% longer training time for each task, MFAN is observed to have about 10% less error rate. These observations call for further theoretical exploration to understand the number of required data points during transitions between different data streams for accurate predictions.

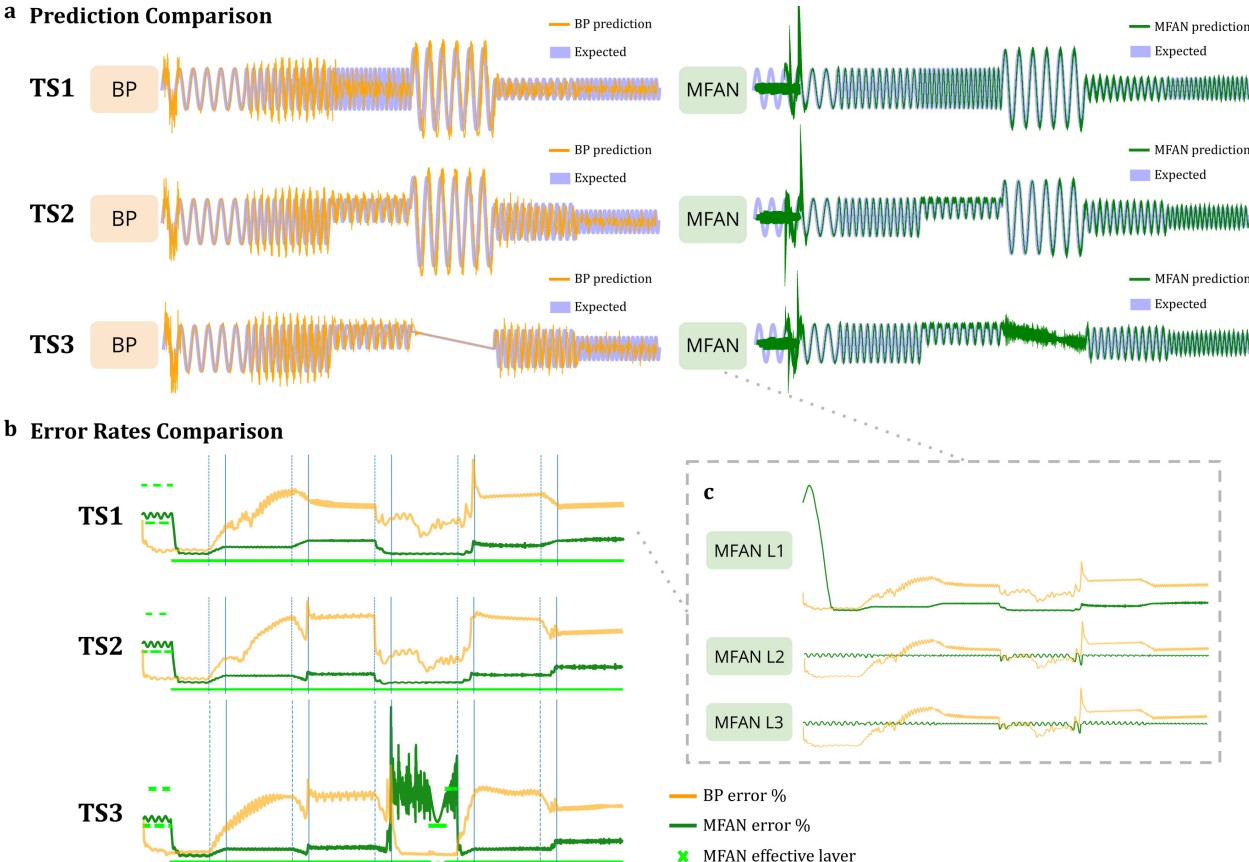

Figure 4: **a)** Predictions for all task streams: TS1 tests the models' ability to handle task transitions and adapt to unseen data. TS2 introduces more complex trigonometric functions, testing the models' ability to deal with increased functional complexity. TS3 introduces a linear function amid trigonometric tasks, testing the models' adaptability during and after a significant change in task distribution. **b)** The percentage error rates corresponding to the predictions in (a), numerically compare the performance of BP and MFAN for each task stream. **c)** Per-layer error rates: The per-layer error rate for MFAN and BP in TS1, illustrates how MFAN dynamically selects layers during training to stabilise error and improve prediction accuracy.

## 4 Discussion

MFAN presents advancements in neural network training, particularly for applications that require continuous data streams and adaptability in dynamic environments. Through integrating contrastive and a multi-layer learning algorithm with dynamic layer adjustment, MFAN offers a promising alternative to traditional BP, demonstrating good stability, performance, and flexibility, especially in handling continuous data of given tasks.

MFAN excels in function regression, where it consistently outperformed BP by capturing complex, nonlinear relationships and generalising effectively to unseen data. The contrastive learning properties (Chen et al., 2020; Du et al., 2024) at the core of MFAN could be one of the primary reasons behind generalizability, while the layer adjustment and independent layer learning characteristics inherited from FF also contribute to better performance.

MFAN's performance in continuous data stream tasks further highlights its adaptability and stability. In the long-term scenario with evolving data, MFAN generated more accurate and stable predictions compared to BP. Moreover, MFAN effectively adapts to varying task complexity and exhibits recovery – the ability to regain performance when transitioning between distinct data phases, including reverting to previously

encountered patterns. This aspect needs further investigation and discussion to potentially support the biological plausibility of MFAN.

### 4.1 Optimisation and Computation: Balancing Complexity and Efficiency

MFAN's flexibility in learning rate adjustment and layer-wise optimisation allows independent tuning of components – $f_x$, $f_y$, and $f_y^{-1}$ – making it responsive to varying data complexities. This adaptability enhances its performance on diverse tasks, though it introduces additional hyper-parameters that require tuning.

MFAN requires about 1.5 times more computational resources than BP due to its complex architecture and takes about 1.2 to 5 times more training time than a single BP, depending on the task. Yet MFAN can achieve half or a third of the percentage error rate in regression tasks than BP, and achieve stability and high accuracy in continuous data stream tasks where BP fails to deliver accurate prediction. Thus, this investment is justified by these substantial advantages in adaptability and predictive accuracy in dynamic settings.

### 4.2 Biological Plausibility

We would like to discuss biological plausibility with the following aspects (Lv et al., 2024) : (i) asymmetry of weights; (ii) local error representation; (iii) non-parallel training, and (iv) the role of recurrent dynamics.

- **Asymmetry of weights:** In traditional artificial neural networks, synaptic weights in the forward path are mirrored exactly along the feedback path through a mechanism called weight transport, essential to back-propagation. This symmetry enables precise error correction but is considered biologically implausible because real neurons are unlikely to share exact synaptic weights. Instead, studies like (Nøkland, 2016) propose methods such as feedback alignment and direct feedback alignment, which replace symmetric weights with random fixed weights, allowing networks to learn effective error-driven representations without strict symmetry.

  The MFAN model addresses this limitation by incorporating localised feedback mechanisms. Instead of relying on global feedback paths, MFAN uses per-layer feedback updates to adjust weights based on local input-output correlations. This design evokes findings in wide neural networks, where weight updates can align with local statistics of inputs (Boopathy & Fiete, 2022). Furthermore, localised feedback reflects the role of recurrent dynamics in the brain, where neural activity depends on both feedforward and recurrent interactions to adapt to complex tasks (Mante et al., 2013).

- **Local error representation:** Biological synapses are thought to adjust their strength using only local information, contrasting with the global error signal used in back-propagation. This local adaptation mirrors how neurons in the brain rely on context-dependent recurrent dynamics to selectively process and integrate sensory inputs (Mante et al., 2013).

  Empirical studies, such as (Ito et al., 2022), demonstrate that task-specific neural circuits dynamically adjust activations using intrinsic connectivity patterns. MFAN's emphasis on localised feedback aligns with these findings. By utilising layerwise adaptation, MFAN enhances its robustness and ability to generalise across tasks.

- **Non-parallel training:** Conventional neural network training often involves distinct forward and backward propagation phases. Instead, natural learning processes are more continuous, with overlapping phases for input integration and weight adaptation. Recent studies propose that biological networks achieve efficient learning through mechanisms like contrastive Hebbian learning and recurrent attractor dynamics (Kymn et al., 2024).

  MFAN excels in handling continuous data streams, demonstrating improvements over traditional BP. Its architecture supports real-time adaptation by updating weights as new data arrives, instead of waiting for batch processing. This dynamic also aligns with findings in the hippocampal-entorhinal system, where modular attractor networks enable compositional representations of spatial and contextual information (Kymn et al., 2024). MFAN also employs contrastive learning to enhance feature

differentiation, which strengthens relevant connections while weakening less significant ones. This mechanism can help to manage the trade-off between retaining prior knowledge and adapting to new inputs.

- **Recurrent dynamics:** While the forward path is often emphasised in biologically plausible models, recent studies highlight the importance of recurrent dynamics for implementing complex behaviours (Mante et al., 2013). Recurrent circuits in the prefrontal cortex enable dynamic selection and integration of task-relevant inputs, as demonstrated by (Mante et al., 2013). Similarly, (Kim et al., 2017) reveals hierarchical cortical organization driven by interneuron distributions, suggesting that recurrent interactions are essential for efficient neural computation. By integrating these principles, MFAN could be extended to recurrent-like processes to enhance learning efficiency and biological plausibility.

MFAN approach lacks some other biologically plausible features, like the use of spikes and unsigned error signals. Spiking neurons enable temporal dynamics and energy efficiency in natural learning processes, while unsigned error signals offer a direction-neutral approach to synaptic adjustments. Fortunately, MFAN's layer-wise-independent design provides flexibility for re-architecture. Each layer acts as a self-contained unit with its own input and feedback loops, making it adaptable to integrating additional components and performing localised adjustments – such as incorporating spiking neural networks (Subbulakshmi Radhakrishnan et al., 2021; Yin et al., 2023), or configuring specific layers for magnitude-only gradient updates – to enhance biological realism.

We acknowledge that deriving formal theoretical guarantees for the alignment of $z_x$ and $z_{y_{\text{pos}}}$ under all possible conditions is a challenging endeavour. However, this work establishes a robust empirical foundation for the assumption. We examined the assumed cosine similarity value after MFAN training and observed high-dimensional vectors with the average element value of 0.9981, close to 1, showing that the alignment between $z_x$ and $z_{y_{\text{pos}}}$ empirically holds. Future work could explore tighter bounds or additional theoretical frameworks to further strengthen this connection.

## 5 Conclusion

In this work, we introduced the Metamorphic Forward Adaptation Network (MFAN), an innovative neural architecture designed to overcome the limitations of traditional back-propagation and Forward-Forward algorithms. MFAN demonstrates flexibility and robustness in tasks requiring adaptation in streaming data, particularly through its dynamic layer depth adjustment and contrastive learning mechanisms. It outperforms baseline models in function regression and continuous data stream by delivering stable and accurate predictions even in complex, evolving environments.

Beyond its technical advancements, MFAN represents a step toward more biologically plausible neural networks. Unlike conventional models relying on global error signals, MFAN employs a localised, layer-wise learning approach akin to how biological systems may process and adapt to information. This allows MFAN to dynamically adjust to changing data streams, aiming to reflect biological learning to reorganise its neural connections in response to new experiences.

Looking ahead, future research could further explore the integration of additional biologically inspired features, such as spiking neurons or more advanced forms of neural plasticity, to further align MFAN with the adaptive learning processes observed in natural systems.

### Acknowledgments

We thank Mohamad Abdul Hady for the discussions during the development phase of this work. This research is partially supported by seedcorn funds by Civil, Aerospace and Design Engineering, Isambard AI and Bristol Digital Futures Institute at the University of Bristol. This work was also partially supported by funding from the Engineering and Physical Sciences Research Council (award no. EP/N018494/1, EP/R026173/1, EP/R009953/1, EP/S031464/1, EP/W001136/1), the SERI funded ERC Proteus consolidator grant (grant no: MB22.00066), the EU H2020 AeroTwin project (grant ID 810321), the Empa-funded DroneHub project,

and the Empa-Imperial College London research partnership. Mirko Kovac was supported by the Royal Society Wolfson fellowship (RSWF/R1/18003).

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

# A  Appendix

## A.1  Classification

### A.1.1  MNIST Classification

Models were trained on 60,000 MNIST images and tested on 10,000 images. Model configurations are outlined in Table A1 below. Accuracy and latent space distribution were analysed.

Table A1: Model setups for classification comparisons.

| MNIST Model Setup | | | |
|---|---|---|---|
| | **MFAN** | **BP** | **FF** |
| **Network depth** | 3-layer | 2-layer | 3-layer |
| **Network width** | $[500, 300, 300]$ | $[500, 300]$ | $[500, 300, 300]$ |
| **Learning rate** | $5e^{-5}, 5e^{-4}, 5e^{-2}$ | $5e^{-4}$ | $5e^{-4}$ |
| $[a_1, a_2]$ | $[0.6, 0.6]$ | / | / |
| **Number of train data** | 60000 | 60000 | 60000 |
| **Number of test data** | 10000 | 10000 | 10000 |
| **Encoder loss** | Cosine Similarity | / | / |
| **Decoder loss** | Cross Entropy | Cross Entropy | Cross Entropy |

### A.1.2  CIFAR-10 Classification

Models were trained on 50,000 CIFAR-10 images and tested on 10,000 images. Model configurations are outlined in Table A2 below. Accuracy and latent space distribution were analysed.

Table A2: Model setups for classification comparisons.

| CIFAR Model Setup | | | |
|---|---|---|---|
| | **MFAN** | **BP** | **FF** |
| **Network depth** | 3-layer | 3-layer | 3-layer |
| **Network width** | $[256, 128, 64]$ | $[256, 128, 64]$ | $[256, 128, 64]$ |
| **Learning rate** | $5e^{-2}, 5e^{-3}, 5e^{-2}$ | $5e^{-3}$ | $5e^{-3}$ |
| $[a_1, a_2]$ | $[0.6, 1]$ | / | / |
| **Number of train data** | 50000 | 50000 | 50000 |
| **Number of test data** | 10000 | 10000 | 10000 |
| **Encoder loss** | Cosine Similarity | / | / |
| **Decoder loss** | Cross Entropy | Cross Entropy | Cross Entropy |

### A.2  Function Regression

Models were trained and tested on self-generated function regression datasets. Models were assessed on generalizability, the understanding of underlying relationships, and the ability to handle functions with various complexities. Model setups are listed in Table A3.

Table A3: Model setups for function regression comparisons.

| FR Model Setup | | |
|---|---|---|
| | **MFAN** | **BP Model** |
| **Network depth** | 6-layer | 3-layer |
| **Network width** | $[128, 128, 64, 64, 32, 32]$ | $[128, 64, 32]$ |
| **Learning rate** | $1e^{-5}$ , $1e^{-4}$ , $1e^{-2}$ | $1e^{-5}$ |
| $[a_1, a_2]$ | $[0.6, 1]$ | / |
| **Number of data** | 100 | 100 |
| **Length of data** | 100 | 100 |
| $x$ **examples:** $x_1$, $x_2$ | $[0, 1, ..., 100], [0.1, 1.1, ..., 100.1]$ | $[0, 1, ..., 100], [0.1, 1.1, ..., 100.1]$ |
| **Train $x$ domain** | $[0, 110]$ | $[0, 110]$ |
| **Test $x$ domain** | $[100, 210]$ | $[100, 210]$ |
| **Range $y_{neg}$** | $[y_{\text{pos}_{\min}} - 0.1, y_{\text{pos}_{\min}} - 0.03]$ $\cup [y_{\text{pos}_{\min}} + 0.03, y_{\text{pos}_{\max}} + 0.1]$ | / |
| **Encoder loss** | Cosine Similarity | / |
| **Decoder loss** | Root Mean Square | Root Mean Square |

### A.2.1 Additional evaluations

Table A4: Additional evaluations.

| Evaluation Aspect | MFAN | BP Model |
|---|---|---|
| **Training Time (3 layers)** | 2.7s | 0.5s |
| **Training Time (6 layers)** | 5s | 1s |
| **Performance Change (3 → 6 layers)** | / | Slight drop
+2% error rate increase |
| **Performance Change (6 → 3 layers)** | Negligible
+0.12% error rate increase | / |
| **Inference Time** | $0.03 - 0.06s$ | $0.02 - 0.04s$ |

## A.3 Continuous Data Stream

Models were trained and tested on self-generated function regression datasets. Models were assessed on generalisability, the understanding of underlying relationships, and the ability to handle functions with various complexities. The model setup is listed in Table A3. Models were evaluated by performing learning for a continuous data stream of three tasks in dynamic environments. The models' predictions and error rates were monitored. Models' long-term dynamic adaptability and stability were evaluated. Detailed model configurations are shown in Table A5.

Table A5: Model setups for function regression with continuous data stream comparisons.

| Model Setup | | |
|---|---|---|
| | **MFAN** | **BP Model** |
| **Network depth** | 3-layer | 3-layer |
| **Network width** | $[64, 32, 16]$ | $[64, 32, 16]$ |
| **Learning rate** | $1e^{-5}, 1e^{-5}, 1e^{-4}$ | $1e^{-1}$ |
| $[a_1, a_2]$ | $[0.6, 1]$ | / |
| **Number of data** | 2901 | 2901 |
| **Length of data** | 100 | 100 |
| $x$ **examples:** $x_1$, $x_2$ | $[0, 0.2, ..., 18.7], [0.2, 0.4, ..., 18.9]$ | $[0, 0.2, ..., 18.7], [0.2, 0.4, ..., 18.9]$ |
| **Range** $y_{neg}$ | $[y_{\text{pos}_{\min}} - 0.1, y_{\text{pos}_{\min}} - 0.03]$ $\cup [y_{\text{pos}_{\min}} + 0.03, y_{\text{pos}_{\max}} + 0.1]$ | / |
| **Encoder loss** | Cosine Similarity | / |
| **Decoder loss** | Root Mean Square | Root Mean Square |
| **Training time** | 0.0034s | 0.0006s |
| **Inference time** | 0.0024s | 0.0004s |

