# OpenReview forum: "Metamorphic Forward Adaptation Network: Dynamically Adaptive and Modular Multi-layer Learning"
_TMLR — Accepted by TMLR_

### Review · Reviewer_1Acp · 2025-01-01

**Summary Of Contributions:**

This paper introduces the Metamorphic Forward Adaptive Network (MFAN), a novel neural network architecture and algorithm inspired by the Forward-Forward (FF) algorithm and contrastive learning. MFAN aims to address limitations of both back-propagation (BP) and FF, particularly in continual learning scenarios. MFAN encodes both inputs and outputs to a latent space, encouraging training input samples to be close to correct outputs and far from incorrect outputs. Then, a decoding network is learned from the latent space to classification or regression outputs to predict on new inputs. Experiments comparing to BP in classification, regression, and continual learning are provided.

**Audience:**

Yes

**Broader Impact Concerns:**

No broader impact statement included: I don't think one is necessary.

**Claims And Evidence:**

No

**Requested Changes:**

Upon making the following changes (as well as answering the questions above), I will re-evaluate this work to determine whether I can recommend it for acceptance.

Your paper should explain your proposed algorithm clearly enough so that a reader could reimplement it themselves. So far, you have only described the loss function without explaining what "training" actually looks like, whether it's back-propagation or not. I recommend including a pseudocode algorithm as well as a much more thorough textual description. This is critical to support your claims in the rest of the paper.

You discuss the FF model quite in-depth in the abstract, introduction, and Figure 1, but only experimentally compare to BP. You should include the original FF algorithm as well as any relevant improvements in your experiments for proper empirical comparison.

How did you select hyperparameters? Improperly selected hyperparameters can confound experimental results by putting one algorithm at an unfair advantage or disadvantage to others. Even further, you should discuss the computational cost of hyperparameter optimization for each approach. Also, why is it fair in your Function Regression experiments that the MFAN model has twice as many layers as the BP model?

Please use `\citep{}` for parenthetical citations. The vast majority of your citations, as currently written, should be parenthetical rather than textual. For example, "...maintaining the plasticity Dohare et al. (2024)." -> ""...maintaining the plasticity (Dohare et al., 2024)."

Generally, the figures and tables appeared quite late compared to when they were referenced in the text. The top of the following page is a good default location.

Also, most figures contain text that is much too small to read at a standard level of zoom: all text should be legible when printed and no smaller than subscripted text in an equation (such as Equation 6).

The captions of figures and tables should only include clarifying information relevant within the figure/table. Any supporting information, such as how the experiments are designed under Figures 2-3, and analysis or discussion, such as the notes under Table 1, should be in the main text.

Specific corrections and changes:

Page 1, abstract: "However, back-propagation is gradient-dependent...": this is redundant after the previous sentence. Rather, why is it bad to be gradient-dependent?

Page 1, abstract: "...and lacks robustness..." -> "...and lacking robustness..."

Page 1, abstract: "A promising alternative to back-propagation is the Forward-Forward algorithm...": either cite this and/or say that it was "previously proposed".

Page 1, abstract: "...not yet matched..." -> "...doesn't yet match..."

Page 1, abstract: "...using contrasting learning property..." -> "...using a contrasting learning property..."

Page 1, introduction paragraph 1: "...advancements in the fields..." -> "...advancements in fields..."

Page 1, introduction paragraph 2: "BP often requires ... continuously changing.": This sentence is not very clear, particularly how the first part is related to the second part. Are you referring to catastrophic forgetting, and/or how the gradient change from a batch of training data may not be a useful step for test data?

Page 1, introduction paragraph 2: missing period at end of last sentence.

Page 2, paragraph 2: "logistic function, $\sigma$ to the goodness minus $\theta$," -> "logistic function, $\sigma$, to the difference of the goodness and $\theta$"

Page 2, paragraph 6: "both x and y as inputs": what are $x$ and $y$? Always define variables, even if they seem obvious.

Page 3, last paragraph: please also provide these results in a table for clarity.

Page 4, Figure 1 caption: letter labeling is inconsistent.

Page 5, Table 1 captions: "The corresponding... Fig. 2)" is a sentence fragment.

Page 6, Figure 2 caption: Two closing parentheses are missing, one in the mathematical expression in the 2nd sentence and the other in the 3rd. Also, I don't understand this "sliding window" approach: it seems different to that described under Figure 3. Particularly, I think these expressions are not correct: "training dataset $x: x_{i+1} = x_i + 0.1i$ with $x_0 = 0$ $i \in \mathbb{Z}$, $i \in [1, 100]$, training dataset $y_{pos} = f(x)$", as they imply $x_0=0, x_1=x_0+0.1\cdot1=0.1, x_2=x_1+0.1\cdot2=0.3, x_3=x_2+0.1\cdot3=0.6, ...$.

Page 8, Figure 3 caption: "Evenly divide the task stream into..." -> "The task stream is evenly divided into...". Also, use `\text{}` to write out words within mathematical expressions: i.e. `"on $\text{segment}_1$ to $\text{segment}_5$,"` yields "on $\text{segment}_1$ to $\text{segment}_5$,".

Page 9, paragraph 1: "While MFAN training demands more computational resources than BP due to its complex architecture, ...": how much more? Please report this for at least one of your experiments.

Page 9, Section 3.2 header: "Biological Plausability" -> "Biological Plausibility"

Pages 12-14: Ensure the appendix tables are labeled and captioned, like "Table A.1: Hyperparameter configurations for MNIST Classification experiments.".

**Strengths And Weaknesses:**

The proposed algorithm seems to be novel and has comparable performance in the presented experiments. However, I cannot fully evaluate the validity of this work until the algorithm and experimental set-ups are clarified. Your proposed algorithm is never actually specified. Are you using gradient descent with these loss functions to update weights for each of the three networks? If so, how does it avoid the multiple apparent downfalls that you have provided for standard BP methods, and how is it still "layer-wise" as you claim in Section 3?

---

> ### Author Response · Authors · 2025-01-25
>
> Thank you very much for the comments and all the questions. We are grateful for your efforts in guiding our manuscript to this stage. We are excited to continue working on finalizing the manuscript. Because of the character limitations, we would like to request to refer to the replies in the Supplementary Material.

---

> > ### Comment · Reviewer_1Acp · 2025-01-26
> > **Further comments**
> >
> > Thank you for your thorough response and modifications. I have some further comments.
> >
> > Regarding Line 6 in Algorithm 1, where is $i$ on the right side of the equation? If the terms on the right side are not meant to be superscripted with $i$ (as it is currently written), then the loss is computed with the final output of full encoder and thus the loss and weight update are not "local"/"layer-wise", and the preceding lines of pseudocode do not correctly have these terms already computed.. If they instead should be superscripted with $i$ such that the loss and weight update are in-fact local, this requires revision as well as explanation. The $i$s are also missing in Line 7.
> >
> > The text size in figures was only somewhat fixed. Figure 1(a) and the legends in Figures 1(c) and 2 still have some text that is not legible.

---

> > > ### Author Response · Authors · 2025-01-29
> > >
> > > Thank you for the comment and apologies for including typos in the resubmitted version. We have addressed these items as follows:
> > >
> > > - We have updated Algorithm 1 to explicitly reflect the layer-wise treatment of loss computation and weight updates. Specifically, the terms in Line 6 and Line 7 now consistently include the layer index $i$ to ensure that the loss is computed locally at each layer rather than at the final encoder output.
> > >
> > > - Additionally, in the decoder section, we have revised Line 11 to use $z_x^i$ ​ instead of the final encoder output to reflect our consistent layer-wise approach throughout both the encoding and decoding stages correctly.
> > >
> > > - We have enhanced the legibility of text in Fig. 1 including the text size issues in Figure 1(a) and the legends in Figure 1(c). Thank you very much again, we couldn't spot other issues and we remain open to addressing any further comments or suggestions.

---

> > > > ### Comment · Reviewer_1Acp · 2025-02-11
> > > > **Response to further comments**
> > > >
> > > > Thank you for addressing these final concerns.
> > > >
> > > > You seem to have previously uploaded responses under "Supplementary material", but this removed the zip file you had originally uploaded here. I believe you would intend to have the original content (as well as potentially the current file of responses) here?

---

> > > > > ### Author Response · Authors · 2025-02-12
> > > > >
> > > > > Thank you for reminding the original supplementary files. Yes, we will keep the current file and the supplementary files including the codes. We also plan to provide GitHub and YouTube links for reproducability pending feedback and permission.

---

> > ### Author Response · Authors · 2025-04-21
> > **Initial Replies**
> >
> > Strengths And Weaknesses: We appreciate the feedback and have taken steps to address the concerns regarding the clarity of the algorithm and experimental setup. For the algorithm and the experimental setup:
> >
> > -  We have added a detailed pseudo-code for both the MFAN encoder and decoder \& layer selection to specify and clarify each step of the proposed algorithm. This pseudo-code provides a clearer understanding of how the model operates and the sequence of computations involved.
> > - The primary difference between MFAN and standard BP lies in the gradient computation. Unlike BP, which uses global weight updates, MFAN employs local gradient computations. This enables relatively independent layer-wise learning, with only directional information passed between layers. This design ensures that MFAN remains consistent with the "layer-wise" learning paradigm discussed in Section 3 while addressing some of the known limitations of standard BP methods.
> >
> > To improve clarity, we included detailed descriptions of the experimental setups within the figure captions corresponding to each evaluation. However, recognizing the need for a better reading flow, we restructured each evaluation section into two subsections: "Evaluation Setup" and "Results \& Analysis". We hope that this reorganization ensures all setup-related information is moved from the figure captions into the main text, allowing readers to follow the logic and context of each experiment more clearly.
> >
> > 1. Clarity: We appreciate your feedback and have made significant revisions to address the concerns regarding the clarity of our proposed algorithm and its implementation.
> >
> > To provide a more thorough introduction to the MFAN algorithm, we have added a new section titled "The Proposed Model." This section introduces the model in greater detail, accompanied by pseudo-codes to clearly outline the steps of the algorithms. We believe that these additions could help readers to have sufficient information to reimplement the approach.
> >
> > We have moved the descriptions of the experimental setups from the figure captions into the main text. This change aims to make it easier for the reader to follow and understand the details of how the evaluations were conducted.
> >
> > The paper has been restructured to present evaluations more effectively. Each evaluation is now highlighted in its own subsection, with clear separation between the "Experimental Setup" and "Results \& Analysis" sections. We hope that it would improve the flow and logic of the paper.
> >
> > We included two detailed algorithms in the new "The Proposed Model" section. These algorithms provide a step-by-step explanation of each component of the MFAN approach, addressing the request for a more textual description alongside the pseudocode.
> >
> > We restructured all evaluations into "Evaluation Setup" section and the "Results \& Analysis" section. This segmentation should help logic flow during reading.
> >
> > 2. FF Algorithm: Thank you for pointing this out. We have revised the paper to include a more thorough comparison with the FF model and its relevant improvements. Below are the key updates:
> >
> > In the new section titled "The Proposed Model," we provide a detailed textual description of the improvements MFAN introduces over the FF algorithm. By moving this discussion into the main body of the paper and supplementing it with pseudo-code, we better illustrate how MFAN addresses the limitations of FF. Specifically, we highlight the following key differences:
> >
> > - MFAN introduces improvements in data stream processing that enhance its applicability to tasks that FF cannot handle.
> > - Instead of relying on FF's "goodness measure," each layer in MFAN independently optimizes a contrastive loss based on cosine similarity. This design allows MFAN to process data in a way that is more flexible and robust than FF.
> > - MFAN can predict without requiring knowledge of all possible $y$, enabling it to handle continuous data and perform regression tasks -- capabilities that the FF algorithm lacks.
> >
> > To provide a more comprehensive evaluation, we included the FF model in our classification experiments and empirically compared its performance to MFAN. These comparisons demonstrate the differences in learning mechanisms and performance between the two models.

---

> > > ### Author Response · Authors · 2025-04-21
> > > **Initial Replies - II**
> > >
> > > 3. Hyperparameters: Thank you for raising this important point. We have taken steps to ensure fairness in our experimental setup and to address the concerns regarding hyperparameter selection and model architecture:
> > >
> > > To address the concern about fairness in the Function Regression experiments, we evaluated both MFAN and BP models using the same number of layers and included the results in the appendix. These results demonstrate that the model design choices do not introduce any significant unfairness in terms of training time, inference time, or error rate performance.
> > >
> > > In the main text, we clarify the rationale for using a six-layer MFAN architecture:
> > >
> > > - The six-layer configuration was implemented to explore MFAN’s adaptive layer selection mechanism, which allows it to dynamically shrink or expand its effective depth during inference.
> > > - In contrast, a three-layer configuration limits this mechanism, as it can only shrink the depth but not expand it. This dynamic adjustment is a unique feature of MFAN, which we wanted to highlight in our experiments.
> > > - Despite this, MFAN most frequently uses three layers during inference, similar to BP. This ensures that the comparison remains fair while also showing MFAN’s added flexibility.
> > >
> > > Hyperparameters were selected carefully to ensure fairness across all models and experiments. We used consistent criteria for optimizing hyperparameters across both MFAN and BP to avoid confounding experimental results or introducing an unfair advantage for either approach. We have included details of the hyperparameter optimization process in the appendix, along with the associated computational costs for each approach.
> > >
> > > The results indicate that the six-layer configuration of MFAN does not unfairly impact its performance relative to BP. In terms of training time, inference time, and error rate performance, MFAN shows no concerning advantages stemming from this design choice.
> > >
> > > 4. Parenthetical Citations: Thank you very much for this comment which helped us to correct the citing style with the use of citep.
> > >
> > > 5. Figures and Tables: Thank you for the advice. We moved figures to reasonable positions to give readers a better reading experience, then we adjusted figures ensuring font sizes were readable. We also reviewed figure captions, moved experimental design information back into the main text, and kept only figure-relevant clarifying information.
> > >
> > > 6. Specific Corrections: Thank you very much for all your comments and detailed correction points which significantly helped us to improve our paper.

---

### Review · Reviewer_uh3q · 2025-01-04

**Summary Of Contributions:**

The paper proposes a new forward-forward algorithm, called Metamorphic Forawrd Adaptation Network (MFAN). The authors use contrastive loss by adding noise to make negative samples. The paper shows the superiority of the proposed method compared to 3-layer MLP trained with the standard back-propagation on MNIST and toy trigonometric functions.

**Audience:**

No

**Broader Impact Concerns:**

I do not have any concerns.

**Claims And Evidence:**

No

**Requested Changes:**

- In Section 1, zy_pos -> $z_{y_\text{pos}}$
- In Eq. 8, $f_y^{-1}f_x$ -> $f_y^{-1}(f_x$

- Move Figure 3 on Page 7 as it is referred to on page 5.

- All citation is citet, please change them to citep.
- In Figure 3 $segment_1tosegment_5$ -> $segment_1$ to $segment_5$.

**Strengths And Weaknesses:**

[Strength]

The paper proposes a biologically plausible network using a forward-forward model with a contrastive loss function. The proposed method, MFAN, demonstrates adaptive prediction capabilities and outperforms the baseline in certain metrics.


[Weakness]

**Theoretical Guarantees:**
A key assumption of the model is that $z_x \approx z_{y_{\text{pos}}}​​$. However, there appears to be no theoretical guarantee for this assumption. This lack of justification weakens the model's foundation.

**Vague Details:**
Several crucial aspects of the model lack sufficient explanation. For example, the effective layer selector is a critical component, yet the authors only mention that "the layer with the best performance on the training set is used at inference time" and "the model records the best-performing layer at the previous time step."
- During the online continual learning tasks, how is the performance of the previous time step quantified during inference?
- Does this imply the model references evaluation results for layer selection?
- What criteria define "best performance"? Is it loss, accuracy, or another metric?

**Implementation Disparities:**
It is unfair to compare MFAN (6 layers) with a baseline model that uses only a 3-layer MLP. Additionally, the paper does not provide details about the training and inference times for MFAN relative to standard backpropagation models of similar size.

**Layer Selection and Adaptive Prediction:**
Adaptive prediction is one of advantages of the proposed method, as argued by the authors although it requires to predict from all layers to choose the target. However, further details about the layer selection mechanism are necessary. I strongly recommend measuring inference time and demonstrating the proposed model's superiority compared to plausible baselines, such as a classifier head at each layer in backpropagation models.

**Implementation Details:**
The paper lacks details about the implementation, such as whether non-linear activation functions are used between layers.

**Related Works:**
The discussion on related works is insufficient. Many biologically plausible methods address the limitations of backpropagation, such as Feedback Alignment [6] and Neural Tangent Kernel [1]. These should be acknowledged and compared.

**Sensitivity Analysis:**
How sensitive is the model's performance to the choice of noise degree in experiments? A sensitivity analysis is highly recommended.

**Task Definition and Misalignment:**
Do you have any intuition as to why the BP model in Figure 2 (center) is slightly misaligned with the ground truth? Could this be due to the task definition?


**Continual Learning Claims:**
Although the authors claim the experiments were conducted in an online continual learning setting, the setup appears to involve streaming data rather than true continual learning. To substantiate the claim, the paper should measure forward and backward transfers across tasks. Furthermore, Figure 3 suggests that the test data closely resembles the training data (e.g., periodic functions with slight shifts). This raises questions about the model's generalizability, especially considering its worse performance than the BP model in Section 2.1.

**Analysis of the ‘Injury’ Period:**
The authors should analyze why MFAN struggles during the "injury" period in TS3.

**Neural Plasticity:**
The claim that "MFAN also demonstrated neural plasticity to some extent" lacks supporting evidence. In standard continual learning, balancing plasticity and stability is crucial. Referring to the task as "online continual learning" adds to the confusion.

**Forward Path Emphasis:**
The authors argue that only the forward path is important for biologically plausible networks. However, several studies emphasize the role of recurrent dynamics in the brain [3,5] and provide methods for implementing such dynamics [2,4].

[1] Boopathy, Akhilan, and Ila Fiete. "How to train your wide neural network without backprop: An input-weight alignment perspective." International Conference on Machine Learning. PMLR, 2022.

[2] Ito, Takuya, et al. "Constructing neural network models from brain data reveals representational transformations linked to adaptive behavior." Nature communications 13.1 (2022): 673.

[3] Kim, Yongsoo, et al. "Brain-wide maps reveal stereotyped cell-type-based cortical architecture and subcortical sexual dimorphism." Cell 171.2 (2017): 456-469.

[4] Kymn, Christopher J., et al. "Binding in hippocampal-entorhinal circuits enables compositionality in cognitive maps." NeurIPS. 2024.

[5] Mante, Valerio, et al. "Context-dependent computation by recurrent dynamics in prefrontal cortex." nature 503.7474 (2013): 78-84.

[6] Nøkland, Arild. "Direct feedback alignment provides learning in deep neural networks." Advances in neural information processing systems 29 (2016).

---

> ### Author Response · Authors · 2025-01-25
>
> Thank you very much for the comments and all the questions. We are grateful for your efforts in guiding our manuscript to this stage. We are excited to continue working on finalizing the manuscript. Because of the character limitations, we would like to request to refer to the replies in the Supplementary Material.

---

> > ### Comment · Reviewer_uh3q · 2025-02-01
> >
> > I appreciate the authors for their response. I have a few follow-up questions.
> >
> >
> > **Re: Consequently, the training objective enforces the condition zx ≃ zypos by design***
> >
> > The loss design does not directly mean that the condition is actually held. I partially agree with the author's point of empirical evidence but I believe measuring the actual similarity of $z_x$ and $z_{y_{pos}}$ is necessary to provide the evidence.
> >
> >
> > **Re: Our additional analysis shows that MFAN reduces the error more effectively during longer periods of consistent patterns.**
> >
> > Thank you for conducting the analysis. Could you refer to which part includes the analysis in the revision?

---

> > > ### Author Response · Authors · 2025-02-04
> > >
> > > Thank you for the comment and follow-up questions which we believe helped us to improve the manuscript. For these two points:
> > >
> > >
> > > 1) For the $z_x$ and $z_{y_{\rm pos}}$ condition, we included actual similarity loss to the last paragraph in Discussion (appeared on page 12, the paragraph before the Conclusion). For ease of track, this is the corresponding part:
> > >
> > > "We examined the assumed cosine similarity value after MFAN training and observed high dimensional vectors with the average element value of $0.9981$, close to $1$, showing the alignment between $z_x$ and $z_{y_{\rm pos}}$ empirically holds."
> > >
> > > 2) We included results from our analyses in the last paragraph of Section 3.3.2 (appeared on page 10, the paragraph before the Discussion). For ease of track, this is the corresponding part:
> > >
> > > "Additional analysis reveals that MFAN decreases its error more effectively and becomes more stable during longer periods of consistent patterns. With 30\% shorter data points, MFAN receives less consistent patterns each time, having 2\% to 3\% larger error rate amplitudes with oscillatory behaviour. Then with 20\% longer training time for each task, MFAN is observed to have about 10\% less error rate. These observations call for further theoretical exploration to understand the number of required data points during transitions between different data streams for accurate predictions."

---

> > ### Author Response · Authors · 2025-04-21
> > **Initial Replies**
> >
> > Since we explored that it is possible to include multiple official comments, it would be easier to keep our replies here rather than a Supplementary Material. We are including our initial replies as follows:
> >
> > Theoretical Guarantees: In our paper, we explicitly state the following: Using the designed loss function, the MFAN encoders are jointly trained to obtain mappings from the input space $(x)$ and the output space $(y_{\rm pos})$ to the latent space, where the embeddings $z_x$ and $z_{y_{\rm pos}}$ are aligned. The loss function inherently encourages these embeddings to become similar during training. This ensures that $z_x$ $\simeq$ $z_{y_{\rm pos}}$ holds empirically in the training set.
> >
> > The similarity measure employed in the encoder design relies on the cosine similarity metric. The cosine similarity is defined in Equation (6) by including vectors $z_x$ and $z_{y_{\rm pos}}$. This value approaches 1 as the angle between the two vectors approaches 0, indicating maximum alignment or similarity.
> >
> > The loss function is explicitly designed to minimize the discrepancy between $z_x$ and $z_{y_{\rm pos}}$ pushing cosine similarity towards 1. Consequently, the training objective enforces the condition $z_x$ $\simeq$ $z_{y_{\rm pos}}$ by design.
> >
> > During the training process, the encoders are optimized to minimize the loss function, which also aligns $z_x$ and $z_{y_{\rm pos}}$. Once the encoders are frozen, the learned mappings are expected to generalize to the test set. Empirical evidence from our experiments supports this claim, as the decoder successfully approximates $f_{y}^{-1}$ by $g_{y}$, and reconstructs $y_{pos}$ at test time (i.e. ${\hat{y}}_{test}$ )
> >
> > using $z_x$ (i.e. $z_{x_{{test}}}$):
> >
> >
> >     \hat{y}_{test} = f_{y}^{-1}(f_x(x_{test})) \simeq g_{y}(z_{x_{test}})
> >
> >
> > This alignment ensures that the assumption $z_x$ $\simeq$ $z_{y_{\rm pos}}$ also holds during inference.
> >
> > While a strict theoretical guarantee may not be explicitly proven in our work, the loss function and optimization process provide a strong practical justification for the assumption. The assumption is not arbitrary but a natural outcome of the training objective, as minimizing the loss inherently enforces the alignment between $z_x$ and $z_{y_{\rm pos}}$.
> >
> > We acknowledge that deriving formal theoretical guarantees for the alignment of $z_x$ and $z_{y_{\rm pos}}$ under all possible conditions is a challenging endeavour. However, this work establishes a robust empirical foundation for the assumption. Future work could explore tighter bounds or additional theoretical frameworks to further strengthen this connection. We indicated this at the end of the Discussions in the new version of our manuscript.
> >
> > Vague Details: Thank you for highlighting this issue. We have revised the manuscript to provide a more thorough explanation of the effective layer selector and its role in the model by addressing each of your specific questions:
> >
> > For the effective layer selector:
> >
> > - We added pseudo-code for the layer selector and accompanying text to clarify the steps involved in selecting the effective layer. This provides a more explicit description of how the selector operates during training and inference for both classification and regression tasks.
> >
> > The selector works as follows:
> >
> > - Each layer in MFAN makes a prediction, and the selector determines the effective layer to use for the final output.
> > - For classification tasks, the layer with the best training performance is used during inference. Alternatively, predictions from multiple top-performing layers can be averaged to produce smoother and more stable results.
> > - For regression tasks, the selector identifies the best-performing layer during training and uses a deeper layer at inference time to account for increased task complexity and extrapolation requirements.
> >
> > Performance metric for Layer selection:
> >
> > - The criteria for "best performance" depends on the task: (i) For classification tasks, we use accuracy to evaluate performance; (ii) For regression tasks, we use mean squared error (MSE) as the performance metric.
> >
> > Dynamic layer adjustment achieved by:
> >
> > - Evaluating the performance of each layer at the previous time step (using accuracy for classification or MSE for regression).
> > - Selecting the best-performing layer from the previous time step to use at the subsequent inference time.
> >
> > This dynamic adjustment mechanism allows the model to adapt to changing data distributions and task complexity effectively.

---

> > > ### Author Response · Authors · 2025-04-21
> > > **Initial Replies - II**
> > >
> > > Layer Selection and Adaptive Prediction: Thank you for your valuable feedback. We have revised the manuscript to address your concerns regarding the layer selection mechanism and inference time, as well as the comparison with potential baselines. Below are the steps we took:
> > >
> > > In the newly added "The Proposed Model" section, we included pseudo-code along with textual explanations to clarify the layer selection mechanism. These additions provide a detailed step-by-step description of how the effective layer is selected during inference, ensuring transparency and reproducibility.
> > >
> > > To evaluate the computational overhead of the layer selection process, we included inference time measurements in our evaluations. These measurements demonstrate that all models, including the proposed MFAN, achieve short inference times (in milliseconds) across the current experimental setup and regression tasks. This suggests that the layer selection mechanism does not introduce significant computational delays compared to the baselines, implying the practical feasibility of the proposed method.
> > >
> > > We decided to retain the comparison with the pure backpropagation (BP) baseline in this study. While a baseline incorporating classifier heads at each layer in BP models is an interesting alternative, we consider this an avenue for future work to further investigate the model's performance in more depth.
> > >
> > > Implementation Disparities: Thank you for raising this important point. We have taken steps to ensure fairness in our experimental setup and to address the concerns regarding hyperparameter selection and model architecture:
> > >
> > > To address the concern about fairness in the Function Regression experiments, we evaluated both MFAN and BP models using the same number of layers and included the results in the appendix. These results demonstrate that the model design choices do not introduce any significant unfairness in terms of training time, inference time, or error rate performance.
> > >
> > > In the main text, we clarify the rationale for using a six-layer MFAN architecture:
> > >
> > > - The six-layer configuration was implemented to explore MFAN’s adaptive layer selection mechanism, which allows it to dynamically shrink or expand its effective depth during inference.
> > > - In contrast, a three-layer configuration limits this mechanism, as it can only shrink the depth but not expand it. This dynamic adjustment is a unique feature of MFAN, which we wanted to highlight in our experiments.
> > > - Despite this, MFAN most frequently uses three layers during inference, similar to BP. This ensures that the comparison remains fair while also showcasing MFAN’s added flexibility.
> > >
> > > Hyperparameters were selected carefully to ensure fairness across all models and experiments. We used consistent criteria for optimising hyperparameters across both MFAN and BP to avoid confounding experimental results or introducing an unfair advantage for either approach. We have included details of the hyperparameter optimisation process in the appendix, along with the associated computational costs for each approach.
> > >
> > > The results indicate that the six-layer configuration of MFAN does not unfairly impact its performance relative to BP. In terms of training time, inference time, and error rate performance, MFAN shows no concerning advantages stemming from this design choice.
> > >
> > > Implementation Details: Thank you very much for pointing this out. We added algorithm pseudo-code, detailed implementation steps as well as experimental setups. For this specific question, we indicated ReLU in Algorithm 1 in the new version of our manuscript.
> > >
> > > Related Works: Thank you for your insightful comment. We have revised the related works section to acknowledge the suggested papers and further address biologically plausible methods that aim to address the limitations of backpropagation. Below are the key updates:
> > >
> > > We have reviewed and incorporated a discussion of the suggested methods, such as Feedback Alignment and Neural Tangent Kernel, into the related works section. These methods are acknowledged and interpreted to the best of our knowledge, and their relevance to the broader context of addressing backpropagation limitations is discussed.
> > >
> > > For this study, we have chosen to retain a comparison with the pure backpropagation (BP) baseline. This decision was made to focus on highlighting the specific contributions and advantages of the proposed MFAN model relative to traditional BP methods. While a direct comparison with other biologically plausible approaches is not included in the current version, we recognize its potential value and are open to future additions.
> > >
> > > Should the current version of our related works discussion be deemed insufficient, we are open to revisiting this section. We would be willing to incorporate additional comparisons and resubmit a revised version of the paper, pending feedback and permission.

---

> > > > ### Author Response · Authors · 2025-04-21
> > > > **Initial Replies - III**
> > > >
> > > > Sensitivity Analysis: Thank you for highlighting the importance of a sensitivity analysis. In response to your feedback, we conducted additional experiments to analyze the model’s performance sensitivity to varying degrees of noise. Below are the key findings:
> > > >
> > > > We included a sensitivity analysis in the function regression section to examine the noise resistance of the MFAN model compared to the BP model. This analysis involved adding Gaussian noise during training, with the same small standard deviation value for both models.
> > > >
> > > > MFAN Performance: MFAN exhibited stable performance under noisy conditions, with only about a 2\% increase in the error rate as the noise level increased. Even at higher noise levels, MFAN maintained relatively steady performance, with a maximum error rate increase of about 5\%.
> > > >
> > > > BP Performance: In contrast, the BP model’s error rate increased more under the same conditions. For small noise levels, BP showed a roughly 8\% increase in error rate, and as the noise level increased further (where MFAN still performed relatively steadily), BP’s error rate rose rapidly, leading to poor predictions.
> > > >
> > > > The superior noise tolerance of MFAN can be attributed to its embedding of noise in the model design. Specifically, the model intentionally adds noise in $y_{neg}$ and learns to distinguish it from the ($x$, $y_{pos}$) pairs during training. This embedded noise handling mechanism aids MFAN’s ability to learn robust representations, which seems to make it more resilient to noisy data as compared to BP.
> > > >
> > > > Task Definition and Misalignment:  Thank you for pointing out the slight misalignment of the BP model with the ground truth in Figure 2 (center). We believe this discrepancy can be attributed to the following:
> > > >
> > > > - During training, the BP model may fail to fully understand the intended learning objective, as it can inadvertently pick up additional irrelevant information from the data. This distracts the model from focusing solely on the desired relationship between $x$ and $y$ which is crucial for accurate predictions.
> > > > - This misalignment becomes more evident during inference, particularly when the model is tasked with extrapolation. Since BP may have learned spurious correlations or irrelevant features during training, it struggles to generalize properly to unseen data and produces predictions that are slightly off the mark.
> > > >
> > > > Continual Learning Claims: Thank you for pointing this out. We are in full agreement with the comments and we decided to keep the scope at the continuous data stream rather than continual learning in the new version of our manuscript.
> > > >
> > > > Analysis of the ‘Injury’ Period: Thank you for highlighting this important observation. We have analyzed MFAN's behaviour during the "injury" period in TS3:
> > > >
> > > > During the "injury" period, MFAN appears to attempt to maintain its periodical prediction patterns, which may explain the temporary struggle in adapting to the sudden shift in the data.
> > > >
> > > > Multiple layer adjustments were observed during this period, indicating that MFAN was actively attempting to recognize the "injury." These adjustments reflect MFAN’s mechanism for dynamically adapting its depth as it processes new and unexpected data streams.
> > > >
> > > > Once the task switched back to trigonometric functions, MFAN quickly adapted and demonstrated high accuracy. This showcases MFAN's ability to recover rapidly and maintain strong performance once the data stream stabilizes.
> > > >
> > > > Our additional analysis shows that MFAN reduces the error more effectively during longer periods of consistent patterns. However, this behavior highlights the need for further theoretical exploration of the number of required data points during transitions between different data streams to ensure sufficient and accurate predictions.
> > > >
> > > > Neural Plasticity: Thank you for pointing out. We fully agree with the comment and removed the claims accordingly.
> > > >
> > > > Forward Path Emphasis: Thank you very much for suggesting these papers and we included these in the new version of our manuscript.

---

### Review · Reviewer_P38z · 2025-01-12

**Summary Of Contributions:**

This is a good paper that introduces a simple change to the Forward-Forward (FF) algorithm to improve it. The authors show that on certain simple tasks, such as regression, their method performs better than backpropagation (Table 1). The paper is well-written, and I felt I could understand everything.

**Audience:**

Yes

**Broader Impact Concerns:**

None.

**Claims And Evidence:**

Yes

**Requested Changes:**

This is a good paper, but I think the authors should run at least one more experiment with a task more challenging than MNIST. This would give readers more confidence that MFAN can scale to harder problems.

If the authors include such an experiment, I will vote to accept the paper. MFAN does not need to outperform BP on this more challenging experiment; however, if MFAN performs worse than BP, the authors should explain why they think this is so, and how they think the algorithm can be improved going forward.

**Strengths And Weaknesses:**

The main downside of the paper, as far as I can see, is the extreme simplicity of the experiments. This, in general, would not be a problem, but the authors write in their introduction:

"Even though FF provides a more modular, layer-wise learning approach, it tends to be slower, particularly when applied to large-scale tasks..."

This suggests they aim to improve upon this property. However, the most complicated task considered is MNIST. It would therefore be beneficial to run MFAN on more challenging tasks, such as CIFAR-100.

Additionally, unless I am misunderstanding the authors' intent, I believe this sentence in the introduction is incorrect: "There is little evidence for the backward flow of error signals, which is central to BP, in biological neural systems."

There is ample evidence for the backward flow of error signals in biology—see [here](https://pubmed.ncbi.nlm.nih.gov/37961227/) for a very recent example.

Finally, this sentence: "As BP relies on global gradient updates, BP-trained models often struggle to adapt to dynamic environments or generalize to evolving tasks" is missing a period at the end.

---

> ### Author Response · Authors · 2025-01-25
>
> Thank you for your thoughtful feedback. Below, we address your concerns regarding the simplicity of the experiments and the need for more challenging tasks:
>
> The classification evaluations in our study were designed to assess MFAN’s ability to handle standard classification tasks. While prior research has already extensively tested both FF and BP on classification tasks (with BP consistently outperforming FF), MFAN was developed primarily to address a different limitation of FF -- its unsuitability for regression tasks. Unlike FF, which has not been applied to continuous data streams in prior work to the best of our knowledge, MFAN demonstrates the capability to handle regression tasks in dynamic environments.
>
> However, we also wanted to show that MFAN can perform classification tasks better than FF, even though its performance does not yet match that of BP.
>
> Acknowledging the simplicity of MNIST, we added the CIFAR image classification task to our experiments. CIFAR presents a more challenging evaluation that helps establish a stronger baseline for MFAN’s classification capabilities.
>
> The results from CIFAR provide valuable insights into MFAN’s performance on more complex classification tasks.
>
> Our primary aim remains to highlight MFAN’s unique ability to handle continuous data, particularly in dynamic environments. Our results demonstrate MFAN’s versatility and its ability to outperform FF in classification and address the limitations of FF in doing regression tasks.
>
> Thank you for your insightful review, we corrected the mistakes in our initial version of the manuscript including signposted 	punctuations and other related grammar issues \& typos, and referred to the suggested paper accordingly. Thank you very much, we included a new comparison for the classification task and included our observations accordingly.

---

### Author Response · Authors · 2025-02-25
**To All Reviewers**

Thank you to all the reviewers for your valuable comments and feedback. We have responded to each of your specific questions in your respective reviews.

We appreciate the opportunity to improve our work and are open to any further discussions, feedback, or clarifications. We believe our revisions address the concerns raised, but we are happy to provide more details if needed. Thank you for your time and consideration in reviewing our work.

---

### Decision · Action_Editor_v3hg · 2025-04-17

**Recommendation:** Accept with minor revision

**Comment:**

The paper is technically sound overall.
However, some of the concerns remain.
Particularly, regarding the sensitivity analysis question, the claim about MFAN’s robustness is not explained with enough detail. There might be pattens in the performance results that require additional attention.
"When Gaussian noise is introduced during training, with small standard deviations, MFAN’s error rate increases by only 2%, while BP’s error rate rises by 8%. With a high noise level, MFAN maintained a relatively steady performance."
Please specify the exact standard deviation and what "high noise" means, and what "relatively steady" is.

Please cross-check for possible inconsistencies of presenting additional experiments and results in the rebuttal and the revised manuscript.

MFAN performs substantially worse than BP on CIFAR-10 (Section 3.1.2, newly added in revision). The provided explanation "However, with much wider networks (over 10 times wider than the short-training setup) and longer training times, both FF and MFAN improved significantly, achieving test accuracies exceeding 46%." needs further elaboration. Can we actually see the performance of BP with a 10x wider network? Efficiency aspects need to be discussed too as MFAN with a wider network requires much more time compared to BP.

**Audience:**

The Forward-Forward algorithm has received considerable attention in the community. The proposed improvement supported by sufficient empirical evidence can be a useful update to the state of the art.

**Claims And Evidence:**

The paper presents a novel approach for improving the Forward-Forward algorithm: the Metamorphic Forward Adaptation Network (MFAN). MFAN employs contrastive loss. It can be used with discrete and continuous data.
The authors demonstrate empirically that MFAN can perform well with respect to stability, adaptability, and ability to handle evolving data and tasks requiring dynamic, long-term learning.

The reviewers find the the novelty to be modest, yet the approach and presented empirical results to be promising.
The presentation of the paper has improved considerably through the discussion phase and rebuttal and revisions performed by the authors addressing the raised concerns by the reviewers.
However, the comprehensibility of the experimental study remains to be a weakness. The author need to revise the manuscript paying special attention at the consistency and conclusiveness of the experimental results and completeness of the discussion.

---

> ### Author Response · Authors · 2025-04-22
> **Minor Edits**
>
> We thank the Editor and reviewers for their feedback. In the revised manuscript, we have implemented the requested clarifications and consistency checks while retaining the structure that the reviewers considered an improvement during the discussion phase. The main changes are summarised below.
>
> 1. Robustness and sensitivity analysis (Gaussian‑noise experiments), corresponding changes:
>
> We introduced Gaussian noise with various standard deviation levels (ranging from 0.5 to 20) during training. With small standard deviations (0.5 to 5), MFAN's error rate is observed to increase by around 2\%, while BP’s error rate rises by 8\%. With high noise levels (standard deviation range from 5 to 20), MFAN maintained a relatively steady performance, with a 6\% error rate increase, whereas BP’s error rate rose rapidly, leading to over 400\% error rate increase and poor predictions overtaken by the added noise.
>
> 2. CIFAR‑10 results, network width and efficiency, corresponding changes:
>
> However, with wider network architectures -- keeping the same 3-layer, short-training setup but scaling each layer to $10$ times the number of nodes, from hundreds to thousands -- and longer training times, both FF and MFAN improved, achieving test accuracies reaching 46\%. Nonetheless, their performance is still inferior to that of the BP model, with also 10 times more nodes, which achieved 49\% test accuracy. In all these conditions, we compared them without additional tuning and only increased the network width by adding 10 $\times$ more nodes to each layer. In such a case, we observed overfitting in BP and some performance increases as discussed for FF and MFAN.
>
> It is worth noting that while MFAN and FF models improved their accuracy with 10 times wider layers, they came at the cost of extended training times, leading to a total of approximately 30 minutes for both MFAN and FF models. This made further optimisation impractical and not worthwhile. In contrast, the BP model with 10 times more nodes required only around 2 minutes for training, underscoring the relative time inefficiency of the FF and MFAN models in vision tasks like CIFAR-10 classification as compared to BP. Overall, BP remained the most efficient and accurate model for this classification task.
>
> 3.  Minor edits and proofreading
>
> We re‑checked and corrected all figure references, inconsistent claims, equation numbers and acronyms. Fig. 1 is also updated with an improved version.